

# Microtopographic control on the ground thermal regime in ice wedge polygons

Charles J. Abolt[1,2], Michael H. Young[2], Adam L. Atchley[3], Dylan R. Harp[3]

[1]Department of Geological Sciences, The University of Texas at Austin, Austin, TX, USA
[2]Bureau of Economic Geology, The University of Texas at Austin, Austin, TX, USA
[3]Earth and Environmental Sciences Division, Los Alamos National Laboratory, Los Alamos, NM, USA

*Correspondence to*: Charles J. Abolt (chuck.abolt@beg.utexas.edu)

**Abstract.** The goal of this research is to constrain the influence of ice wedge polygon topography on near-surface ground
temperatures. Because ice wedge polygon topography is prone to rapid change in a changing climate, and because cracking in
the ice wedge depends on thermal conditions at the top of the permafrost, feedbacks between topography and ground
temperature can shed light on the potential for future ice wedge cracking in the Arctic. We first report on a year of subdaily
ground temperature observations at five depths and nine locations throughout a cluster of low-centered polygons near Prudhoe
Bay, AK, and demonstrate that the rims become the coldest zone of the polygon during winter, due to thinner snowpack. We
then calibrate a polygon-scale numerical model of coupled thermal and hydrologic processes against this dataset, achieving an
RMSE of less than 1.2°C between observed and simulated ground temperature. Finally, we conduct a sensitivity analysis of
the model by systematically manipulating the height of the rims and the depth of the troughs, and tracking the effects on ice
wedge temperature. The results indicate that deeper troughs lead to increased snow entrapment, promoting insulation of the
ice wedge. Rims act as preferential outlets of subsurface heat; increasing rim size decreases winter temperatures in the ice
wedge. The potential for ice wedge cracking is therefore reduced if rims are destroyed or if troughs subside, due to warmer
conditions in the ice wedge. These findings can help explain the origins of secondary ice wedges in modern and ancient
polygons. The findings also imply that the potential for reestablishing rims in modern thermokarst-affected terrain will be
precluded by reduced cracking activity in the ice wedges, even if regional air temperatures stabilize.

## 1 Introduction

It has long been understood that the formation of ice wedge polygons is intimately linked with thermal contraction
ground stresses (Leffingwell, 1915; Lachenbruch, 1962; Mackay, 2000). However, changes to the near surface thermal regime
as polygon topography develops are poorly constrained. Across the continuous permafrost zone, winter ground temperatures
below -10°C commonly drive the opening of a network of cracks, ~4-5 m deep, that bound polygonal regions of soil, 10-30 m
in diameter. Over timescales of centuries to millennia, repeated cracking and infilling with ice in the same locations produces
wedge-shaped bodies of ice at the top of the permafrost, up to 5 m wide (Kanevskiy et al., 2013). Particularly in coastal regions



of the Arctic, this slow growth of ice wedges results in subtle but distinctive surface topography, as pressure between the wedge and the adjacent ground creates rims of raised soil at the perimeters of the polygons. Although only on the scale of decimeters, this microtopography profoundly influences tundra hydrology (Liljedahl et al., 2012; 2016), and may exert equally strong controls on microbial conversion of soil organic carbon into carbon dioxide and methane (Zona et al., 2011; Wainwright

et al., 2017). Polygon microtopography also controls depth variation in the winter snowpack, which accumulates preferentially in low zones, such as the trough space between polygons (Wainwright et al., 2017). It is well known that snow accumulation in periglacial terrain strongly controls winter ground temperatures, by providing insulation from the atmosphere (e.g., Mackay and MacKay, 1974; Goodrich, 1982). Moreover, it has been widely observed that changes to polygon topography have accelerated in the past three decades, as rising air temperatures have increasingly driven the subsidence of troughs and the

destruction of rims (Jorgenson et al., 2006; Raynolds et al., 2014; Liljedahl et al., 2016). However, feedbacks between topographic change and the thermal regime of the ice wedge, which directly control the likelihood of sustained ice wedge cracking in the future, are incompletely understood.

In this paper, we quantify the relationship between the microtopography of ice wedge polygons and subsurface temperatures, using a combination of field observations and numerical modeling. We first analyze a high-resolution record of

ground temperature in key locations beneath a low-centered polygon near Prudhoe Bay, Alaska, and use the observational data to calibrate a numerical model of coupled thermal and hydrologic processes at the polygon scale. We then conduct a sensitivity analysis to determine the influence of two topographic attributes, rim height and trough depth, on winter temperatures in the ice wedges. To our knowledge, our analysis is the first to quantify influences on ground temperature associated with these two variables, which are difficult to control in field experiments, and prone to rapid variation as polygons develop and respond to

a changing climate (Jorgenson et al., 2006; Raynolds et al., 2014; Liljedahl et al., 2016). The results shed light on feedbacks that will determine the sustainability of ice wedge cracking in the near future, as warming air temperatures drive rapid surface deformation in polygonal terrain. The results are also relevant to understanding of processes associated with historical development of ice wedge polygons, including the formation of "secondary" ice wedges, or comparatively young wedges that subdivide older polygons, observed in modern and ancient systems (Burn and O'Neill, 2015).

**2 Background**

A substantial amount of research has explored interactions between ground temperatures and ice wedge cracking. The earliest, and still most rigorous, mechanical analysis was conducted by Lachenbruch (1962, 1966), who concluded that the probability of a cracking event is determined by two criteria: whether the ground temperature cools below a variable threshold near -10°C, and whether the rate of cooling is sufficient. These conditions are most crucial at the top of the permafrost, or the

top of a pre-existing ice wedge, as indicated by theory and observations of active ice wedges (Mackay, 1984). Recently, a number of investigations have used electronic temperature sensors to more tightly constrain thermal conditions at the time of cracking at field sites across the Arctic (e.g., Mackay, 1993; Allard and Kasper, 1998; Christiansen, 2005; Fortier and Allard,



2005; Kokelj et al., 2007). Although heterogeneity exists between sites, it has been generalized that ice wedge cracking is most favorable when ground temperature drops below -13°C, and the rate of cooling exceeds 0.1°C d$^{-1}$ for two days or more (Morse and Burn, 2013).

In contrast to research on ice wedge cracking, relatively few investigations have explored systematic variation in ground temperatures associated with polygon topography. In particular, few datasets have been published that monitor ground temperature beneath distinct zones of a polygon in the same season. Nonetheless, one consistently observed trend is that the rims of a low-centered polygon tend to become several degrees colder in winter than the center or troughs (Mackay, 1993; Atchley et al., 2015). This effect is attributed to the thinner snowpack on top of the rims, as wind-driven redistribution of snow enhances accumulation in topographic lows. Consistent with these observations, previous conceptual models of the thermal regime of active layers in ice wedge polygons have incorporated the idea that cooling is enhanced in raised zones, such as rims (Christiansen et al., 2015) and impeded in low ones (Gamon et al., 2012). However, the strength of these effects, and interactions between them at the polygon scale, remain unclear. For example, if the influence of enhanced cooling in the rims of a low-centered polygon extends to adjacent regions of the subsurface, it seems likely that the gradual development of rims should promote colder temperatures in the ice wedge. This effect would represent a positive feedback on development of low-centered polygons, because colder temperatures favor cracking and the expansion of the ice wedge. On the other hand, it has also been suggested that development of relief in the rims drives precisely the opposite effect, by increasing snow entrapment in the troughs, thereby enhancing insulation of the ice wedges (Lachenbruch, 1966).

Improved understanding of interactions between topography, snow depth, and ground temperature is needed to resolve these conflicting conceptual models, because feedbacks between these variables may have important ramifications for ice wedge cracking at all stages of polygon development. For example, it was argued recently that the presence of secondary ice wedges, or young wedges that subdivide older polygons, results from deactivation of the older primary network due to increased snow entrapment in the troughs, either as the rims grow or the trough itself subsides (Burn and O'Neill, 2015). This conceptual model would inform interpretation of both modern wedges and relict Pleistocene-aged ice wedge casts. However, it competes with a second hypothesis, that secondary wedges are instead reflections of infrequent severe winter conditions, during which cracking in the primary network alone is insufficient to relieve thermal contraction stresses (Dostovalov and Popov, 1966). This latter hypothesis was supported by a numerical model of ground cracking under an imposed tensile stress (Plug and Werner, 2002), but the model was criticized for failing to represent heterogeneity in the subsurface stress field associated with microtopography (Burn, 2004).

Feedbacks between topography and subsurface temperatures are likewise directly relevant to conceptualizing modern thermokarst development in the Arctic, as the changing form of polygons may influence the probability of sustained cracking in degraded ice wedges. As permafrost degradation has accelerated in recent decades (Walker et al., 1987; Osterkamp and Romanovsky, 1996; Jorgenson et al. 2006; Raynolds et al., 2014; Liljedahl et al., 2016), thaw in the upper portions of ice wedges has increasingly triggered trough subsidence and the destruction of low-centered polygon rims, creating convex-up, high-centered polygons. Raynolds et al. (2014) condensed years of field observations into a conceptual model of the process,





suggesting that, while the degradation of low-centered polygon rims has historically been a reversible process, much of the recent thermokarst has proceeded to irreversible extents, due to the destruction of an ice-rich "intermediate" or "transition layer" at the top of the permafrost, which normally buffers deeper zones from thaw. Presumably, the permanence (or reversibility) of modern thermokarst will also be determined by the potential for future ice wedge cracking, which will be

necessary to reestablish polygon rims. Understanding of the potential for ice wedges in degraded troughs to cool to temperatures suitable for cracking is therefore important for predicting the duration of changes to landscape-scale processes associated with high-centered polygon development, such as increased runoff, decreased evapotranspiration (Liljedahl et al., 2012; 2016), and increased emissions of carbon dioxide (Wainwright et al., 2015).

## 3 Methods

**3.1 Study area and data acquisition**

       The study site is centered on a low-centered polygon approximately 40 km south of Prudhoe Bay and 1 km west of the Dalton Highway in Alaska's North Slope Borough. The surficial geology of the region is dominated by fluvial and marine-fluvial silty sands, associated with streams flowing north from the Brooks Range (Jorgenson and Shur, 2007; Raynolds et al., 2014). These deposits are capped with approximately 2 m of Pleistocene-aged aeolian silt, which grades upward into a surface

mantle of peat (Everett, 1980). Active layer thickness varies from 70-90 cm, extending into the aeolian silt layer. Vegetation at the site is typical of the region, and consists almost exclusively of low-lying sedges and grasses. Mean annual air temperature from 2000-2015 was -8.9°C, as estimated by Noah land surface model output associated with NASA's Global Land Data Assimilation System (GLDAS) (Rodell, 2004).

       The microtopography of the studied polygon (outlined in green) is represented in a 50 cm resolution lidar digital

elevation model (DEM) (Fig 1). Like many low-centered polygons south of Prudhoe Bay, the polygon has modest relief compared with polygons elsewhere in the Arctic, with surface elevations ranging from ~78.8-79.2 m above sea level. The relatively low rims and the presence of standing water at the eastern vertex of the polygon suggest that some ice wedge degradation has occurred in recent decades, as documented at nearby sites (Raynolds et al. 2014). Although the polygon is not within one, drained thaw lake basins are common in the area, and two moderate-sized thermokarst lakes are less than a km

distant in opposite directions.

       Field work was conducted in late July, 2014 and early September, 2015. During the first visit, temperature "sensor rods" (Alpha Mach, Sainte-Julie, Quebec, Canada) were installed across the polygon (locations shown as blue dots in Fig 1). Each rod was equipped with thermistors embedded in a water-proof plastic pipe, and driven into the active layer to observe temperature at depths of 10, 20, 30, 40, and 50 cm. Each sensor rod was equipped with an onboard power source and data

logger programmed to record temperature every three hours. The temperature resolution of the sensors was 0.125°C, and their accuracy was estimated by the manufacturer at ±0.5°C. Sensor rods were installed into a mix of higher-elevation rim sites, and lower-elevation sites from the polygon interiors. Rods were removed and data were downloaded in 2015. An example of the



data from rod a101 (Fig 2) displays temperature data from September 2014 through August 2015. Figures representing all data rods are included in the Supplemental Information.

In addition to sensor rod installation, soil cores were collected in 2014 and subsequently analyzed for hydraulic and thermal properties. Soil cores were collected in both the center and the rims, and at depths varying from ground surface to 19

cm. Soil cores below 19 cm could not be collected due to a high water table. Laboratory analyses of the soil cores, described in the Supplemental Information, informed the parameterization of our numerical model.

### 3.2 Statistical analysis of observational data

Prior to constructing a numerical model, data from the sensor rods were analyzed to understand differences in the subsurface thermal regime between the rims and other areas of the polygon. The seasonality of these differences was also

evaluated to determine the potential relationship with snow cover. The one-tailed rank sum test was used to evaluate the hypothesis that minimum winter temperature in the rims (n=3, a000, a108, a109) are colder than polygon centers (n=6, all remaining rods) at all observed depths, and to determine whether the autumn freezing curtain is shorter in the rims than elsewhere. Freezing curtain duration is defined at each sensor as the length of time that ground temperature remained between -0.5°C and 0.5°C, or approximately zero within the accuracy of the sensors, due to the release of latent heat as soil water

freezes. We expected that rims would experience shorter freezing curtains, due to enhanced cooling and decreased soil water content. Finally, to analyze seasonality, the rank sum test was used to determine whether mean monthly temperatures are coldest in the rims, each month from September 2014 through August 2015. We expected that rims would only be colder than the center during months with significant snow cover.

### 3.3 Overview of the Advanced Terrestrial Simulator

Our numerical model used the Advanced Terrestrial Simulator (ATS, version 0.86), a code developed by the United States Department of Energy within the Amanzi framework (Moulton et al., 2011) to simulate surface and near-surface thermal hydrology in variably saturated media (https://github.com/amanzi/ats). ATS uses a multi-physics process management tool called Arcos (Coon et al., 2015) to allow for flexibility in coupling self-contained models for each component of the physical system (e.g., the subsurface mass balance and the surface energy balance). ATS was chosen due to its successful prior

application to lowland permafrost terrain (Atchley et al., 2015; 2016; Harp et al., 2016), and for its rich collection of features tailored to simulating hydrologic processes in cold environments. One component of ATS that is particularly useful for our study is a unique module that emulates wind-blown redistribution of snow across topography using a diffusion-wave equation, taking the same mathematical form as the Mannings equation. This causes snow accumulation to vary inversely with elevation, by leveling the top of the snowpack into a flat surface. In the present implementation, we coupled the subsurface conservation

of mass and energy with a surface energy balance model, which was driven by meteorological data and allowed for accumulation of liquid water, ice, and snow at the surface. An overview of the partial differential equations governing surface and subsurface processes is presented in Painter et al. (2016). A detailed explanation of the surface energy balance, including



description of a snowpack aging model used to estimate changing snow thermal conductivity and albedo throughout the winter is presented in Atchley et al. (2015).

### 3.4 Model construction, calibration, and sensitivity analysis

Construction and calibration of the numerical model followed the workflow described by Atchley et al. (2015), with each model run comprising three computer simulations. In the first simulation, a water table was established near the surface in isothermal conditions by defining a constant-pressure (Dirichlet) boundary condition at the bottom of the domain. In the second simulation, permafrost conditions were established by adding a constant-temperature boundary condition at the bottom (50 m depth), allowing the soil column to freeze from below. In the third simulation, the surface energy balance was introduced, employing meteorological data to define transient thermal and hydraulic boundary conditions at the top of the domain.

Due to the availability of field samples used to estimate soil physical properties, and in an effort to avoid over-fitting the model, calibration focused solely on snow pack parameters. As described in the Supplemental Information, the calibrated parameters included the thermal conductivity of fresh snow, the snow redistribution coefficient used to transport snowpack across variable topography, and a snowfall multiplier used to correct for under-reporting in our meteorological forcing data. For calibration, a 2D domain was developed using topography from the lidar DEM, which included four ground materials (Fig 3). The domain extended laterally from approximately 1 m NW of rod a103 to 2 m SE of rod105, intersecting five sensor rods in different topographic positions. The bottom boundary temperature was set at -8°C, characteristic of nearby borehole observations (Clow, 2014). To approximate gradation of surficial peat into mineral soil at our site, the upper 2.5 cm of the soil column was defined as unconsolidated peat, and the next 30 cm as more tightly compacted peat. Soil hydraulic and thermal parameters for these upper layers were assigned from laboratory estimates from the core samples. The lower soil layers were modeled as mineral soils, to which the ATS default parameters, characteristic of a silty sand, were applied. Ice wedges were included beneath the troughs at a depth of 80 cm, consistent with an active layer survey conducted in September, 2015. Physical parameters for all ground materials are summarized in Table 1.

Meteorological variables used to drive simulations included air temperature, wind speed, incident shortwave radiation, longwave radiation, relative humidity, rainfall, and snowfall. Time series of each variable were derived from the output of the Noah Land Surface Model as distributed by GLDAS, which provides estimates of global weather conditions from year 2000 to the present, operating at a spatial resolution of 0.25° and a temporal resolution of 3 hrs (Rodell, 2004). Meteorological data were extracted from the pixel centered at 148.875°W, 69.875°N (approximately 5 km from our field site) for the time period from September 1, 2010-August 31, 2015, and each variable was averaged into daily means. The first four years of this period were used as spin-up, and the fifth year was used to compare simulated against observed temperatures.

After obtaining a suitable calibration using field site topography, a sensitivity analysis was conducted by repeating simulations with hydraulic and thermal parameters held constant, but with the height of the rims and the depth of the troughs being systematically altered. Following each simulation, winter temperatures were extracted from 2014-2015 from the uppermost cell at the center of the SE (right) ice wedge, and compared to the criteria identified by Morse and Burn (2013) that



favor ice wedge cracking. Simulations scanned through 6 different rim heights, varying from -10 cm to +15 cm in increments of 5 cm; and 5 different trough depths, varying from unchanged to 40 cm deeper, in increments of 10 cm. These ranges were chosen to recreate the variability observed near our field site, and to match recent documentation of troughs impacted to various degrees by ice wedge degradation (Jorgenson et al., 2006).

The ensemble of topographies for our analysis (Fig 4) were created by altering the original mesh used to represent field site topography. When either trough depth or rim height was increased, the elevation of every rim or trough node was directly reassigned. Abolt et al. (2017) showed that the progression of polygonal topography from low-centered to high-centered form is closely approximated by the linear hillslope diffusion equation; therefore, when relief was reduced in the rims, a linear diffusion operator was applied to all non-trough nodes until the elevation of the southeast rim decreased by the

desired amount. This procedure reproduced the smooth, convex-up topography expected to develop as the rims of a low-centered polygon degrade.

## 4 Results

### 4.1 Statistical analysis of observational data

     Minimum winter temperatures (Table 2) and freezing curtain durations (Table 3) observed among all sensor rods

showed considerable variability, with temperatures in the rims becoming colder and falling below 0°C sooner than those in the centers. The results of the rank sum tests (Table 4) confirm that the difference between minimum winter temperature in the rims and in the centers is significant at all depths ($p < 0.025$, indicating a low probability that variations could be explained by random processes), with median differences varying from 3.2°C at 10 cm to 2.3°C at 50 cm depth. Similarly, freezing curtains are shorter in the rims than the centers at all observed depths ($p < 0.025$), the median difference being approximately 10 days.

The results of the rank sum tests, evaluating the hypothesis that rims experience mean monthly temperatures colder than the centers, reveal a stark seasonal pattern in which rims are significantly colder than the center ($p < 0.1$) only during the winter (Fig 5). The difference first becomes significant at a depth of 10 cm in October, but requires an additional two months to become significant at a depth of 50 cm. Rims remain significantly colder than centers through the month of March, after which there is no significant difference through the period of observation.

### 4.2 Model calibration

     Results of model calibration indicate that the best-performing snowfall multiplier was 1.7. Using this value, maximum snow depth in the center of the polygon during the winter of 2014-2015 was approximately 45 cm, comparing well with ground observations from SNOTEL stations at Deadhorse and Sagwon (approximately 40 km north and 40 km south of the study site, respectively; data available at https://www.wcc.nrcs.usda.gov/snow/). The optimal thermal conductivity for freshly fallen snow

was 0.019 W m$^{-1}$ K$^{-1}$, which is within the range of recently published field measurements (Riche and Scheebeli, 2013; Domine et al., 2016). Additionally, the snow redistribution coefficient was reduced an order of magnitude from the ATS default value,





effectively increasing the speed with which the snowpack developed a level surface in winter. Using these parameters, RMSE between simulated and observed temperature from the year of observation, incorporating all twenty-five sensors embedded in the 2D transect (5 thermistors in each of 5 sensor rods), was approximately 1.1°C. RMSE at individual rods varied from ~1.3°C at rod a109 to ~0.8°C at rod a103. Plots comparing simulated and observed ground temperature at rod a101 (in the center) and

at rod a109 (on a rim) demonstrate a close visual match (Fig 6). A snapshot of simulated ground temperature and snowpack on December 24, 2014 (Fig 7) clearly illustrates zonation in the ground thermal regime, whereby the rims become the coldest zone of the polygon.

### 4.3 Sensitivity analysis

The criteria we used to determine favorable conditions for ice wedge cracking were whether winter 2014-2015

temperatures at the top of the southeast ice wedge cooled below -13°C, and whether the rate of cooling surpassed 0.1°C d$^{-1}$ for two days or more (Morse and Burn, 2013). The range of topographies simulated in our sensitivity analysis straddled these conditions (Table 5), with minimum temperatures at the top of the ice wedge varying from -16.5°C in a polygon with rims 15 cm higher and a trough the same depth as our field site, to -12.4°C in a polygon with rims 10 cm lower and a trough 40 cm deeper. In all cases in which temperature cooled below -13°C, the rate of cooling was sufficient to make cracking favorable.

Minimum winter temperature in the ice wedge consistently increased with trough depth, and decreased with rim height. Cracking was determined to be favorable in most simulations, as the ice wedge failed to cool below -13°C only when rim height was equal to or less than our field site, and trough depth was greater.

## 5 Discussion

### 5.1 Zonation in the subsurface thermal regime at our field site

Data from our field site demonstrated a clear pattern in which low-centered polygon rims become the coldest region of the subsurface in winter, even in a polygon with relatively modest relief. The stark seasonality of this pattern, whereby rims become colder than the centers only after snow has accumulated on the ground, is consistent with the hypothesis that most of the variation in subsurface temperatures can be explained by snow depth variation associated with microtopography. Although this finding was expected, our 2D model using field site topography is, to our knowledge, the first physical simulation at the

polygon scale to demonstrate that depth variation induced by leveling of the snowpack surface is sufficient to explain observed zonation in the subsurface thermal regime. Our confidence in the model is reinforced by the low RMSE between observed and simulated temperatures, obtained using soil physical parameters derived from core samples, and a calibrated estimate of snow thermal conductivity that fits recent field measurements (Riche and Scheebeli, 2013; Domine et al., 2016).

The findings from our rank sum tests, that even relatively small rims become colder than the polygon centers in

winter, and that rims fully freeze before the rest of the polygon, support the hypothesis that rims are sites of enhanced cooling in low-centered polygons. Moreover, our simulation results imply that the effects of enhanced cooling in the rims extend to



adjacent regions of the subsurface, as temperature gradients throughout the active layer in early winter favor the transfer of heat from the center and the troughs toward the rims (Fig 7). Complementing this trend, a number of physical factors underscore the potential for rims to act as preferential outlets of subsurface energy. Because the rims are the first region of the polygon to experience sub-freezing temperatures, laterally-oriented thermal gradients are established very early in winter, just as large quantities of latent heat are released by the phase change of soil water (from liquid to solid) in the centers and troughs. At precisely the same time, the thermal conductivity of soil in the rims also increases abruptly due to freezing, increasing the potential for preferential transmittance of heat. Working synergistically, these factors can deliver a considerable boost to freezing and cooling processes in subsurface regions adjacent to the rim, including the ice wedge.

## 5.2 Sensitivity of ice wedge temperatures to topography

The key insight delivered by our sensitivity analysis is that both trough depth and rim height have substantial influence on wintertime temperatures in the ice wedge. Unsurprisingly, deeper troughs are associated with warmer ice wedge temperatures, due to increased insulation of the active layer directly above the wedge. Perhaps less intuitively, however, rim height appears to hold even greater influence on the thermal regime of the ice wedge: on average, varying rim height across a 25 cm range produced a 3.2°C change in minimum ice wedge temperature, compared with 1.0°C of variation associated with a 40 cm range of trough depth. Rims in our simulations consistently acted as preferential outlets of energy from the subsurface in winter, because in all cases, increased rim heights were associated with colder ice wedge temperatures, due to lateral heat conduction from the troughs. The relatively high sensitivity of ice wedge temperature to rim height implies that most of the cooling experienced by the ice wedge in winter occurs through the rims, rather than through the active layer of the troughs. Moreover, the proportion of cooling attributable to the rim increases with rim size, because larger rims decrease the sensitivity of ice wedge temperature to trough depth (Table 5).

The high sensitivity of ice wedge temperature to rim height has not been emphasized in previous conceptual models of permafrost dynamics in polygonal terrain; however, our analysis implies it has a crucial function in feedbacks associated with polygon development, as it can make the difference between whether or not an ice wedge cracks in any given winter. For example, in the final year of our simulations, the ice wedge did not cool below -13°C in a polygon with rims equal to or smaller than our field site and a trough 40 cm lower, representing an advanced stage of thermokarst development. On the other hand, the ice wedge cools considerably below -13°C in a polygon that has equally degraded troughs, but has rims 15 cm higher than the field site, a form resembling the enigmatic "fortress polygons" described by Root (1978) and Mackay (2000). This finding implies that ice wedge cracking is far more favorable, and therefore more frequent, in fortress polygons as compared with high-centered polygons, despite that both forms represent thermokarst trajectories (Mackay, 2000). Moreover, this mechanism could help explain the persistence of low-centered form in fortress polygons, which would be reinforced by sustained ice wedge growth.

The role of rims as preferential outlets of subsurface heat is consistent with the idea that topographic highs in ice wedge polygons cool more efficiently than depressed areas (Christiansen, 2005; Gamon et al., 2012), but conflicts with the





idea previously suggested by Lachenbruch (1966), that rim development gradually suppresses ice wedge cracking by increasing snow entrapment in the troughs. It is important to acknowledge that our model does not fully negate Lachenbruch's hypothesis, as conditions not incorporated into the sensitivity analysis, such as vegetation differences between the rims and center, can also influence snow accumulation patterns and energy exchange between the ground and atmosphere (Gamon et al., 2012). Nonetheless, our results strongly suggest that, considered as an independent variable, increased rim height enhances wintertime cooling in the ice wedge.

Overall, our analysis strongly affirms the idea that topography of ice wedge polygons drives considerable and systematic spatial variation in the subsurface thermal regime, which must be considered in any conceptualization of ground thermal contraction stresses (Burn, 2004; Burn and O'Neill, 2015; Lachenbruch, 1962; 1966). Regarding historical polygon development, the results provide evidence that feedbacks associated with microtopographic change are sufficient to explain the presence of secondary wedges in modern and ancient polygons, as rim destruction and trough deepening are common events on the tundra, and both suppress the potential for cracking. If the primary ice wedges surrounding a polygon are deactivated through these mechanisms, a secondary wedge may form so long as thermal conditions in the center of the polygon remain favorable for cracking. The new wedge would be initiated by contraction stresses which had been relieved in previous winters through activity in the primary network. Cracking in the new wedge would become increasingly frequent if rims begin to develop adjacent to it, or if the troughs above the primary network continue to subside.

This same concept, that low rims and deep troughs suppress ice wedge cracking, also has important implications regarding the permanence of recent thermokarst development across the Arctic (e.g., Jorgenson et al., 2006; Liljedahl et al., 2016). It has already been emphasized that the destruction of an ice-rich transition layer at the top of the permafrost may render irreversible much of the regional-scale thermokarst observed in the past three decades (Raynolds et al., 2014). Our model indicates that changes to topography associated with ice wedge thaw compound this effect, as the development of high-centered polygon topography impedes the ability of the ice wedge to cool to temperatures favorable for cracking. Because of this topographic disadvantage, the ice wedges surrounding high-centered polygons should crack infrequently, and might only return to normal levels of activity if the winter climate becomes colder (or less snowy) than the conditions in which the wedges first formed. Thus, the re-establishment of rims in thermokarst terrain is unlikely should future air temperatures remain on a warming trajectory, or even if the climate stabilizes. This implies that regional scale changes to tundra hydrology (Liljedahl et al., 2012; 2016) and microbial processing of soil organic carbon (Zona et al., 2011; Wainwright et al., 2015) associated with high centered polygon development are likely to persist, once initiated, for timescales mirroring regional climate fluctuations.

**6 Conclusions**

Our analysis of observational data confirms that the microtopography of ice wedge polygons drives considerable, systematic variation in near surface ground temperatures, even in a set of polygons with relatively modest relief. This variation is most notable in winter, as subsurface cooling is most efficient beneath topographic highs, and impeded beneath topographic



lows. Our numerical model reveals that rims act as preferential outlets of subsurface heat in low-centered polygons, because lateral temperature gradients drive energy transfer from adjacent regions of the subsurface toward the rims in winter. Therefore, increased rim size drives colder temperatures in the ice wedge. Rim size and trough depth represent critical factors influencing whether an ice wedge becomes cold enough to crack during winter. Therefore, feedbacks between topographic change and

subsurface temperatures can explain deactivation of the primary network and development of secondary ice wedges in modern and ancient polygons. Because decreased rim size and increased trough depth both suppress ice wedge cracking, development of modern thermokarst topography is likely to reduce rates of ice wedge growth, precluding the reestablishment of rims around degraded troughs.

## Acknowledgements

We are grateful for the support provided for this research, which included: NASA's Jet Propulsion Laboratory under contract #C021199 (Erika Podest is our Project Manager); the NASA Earth and Space Science Fellowship program, for an award to CJA; and the Next Generation Ecosystem Experiments Arctic (NGEE-Arctic) project (DOE ERKP757) funded by the Office of Biological and Environmental Research in the US Department of Energy Office of Science.

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



**Table 1: Ground physical (hydraulic and thermal) properties used in model.**

| | Porosity | Residual water content | Van Genuchten α | Van Genuchten m | Intrinsic permeability | Saturated thermal conductivity | Dry thermal conductivity |
|---|---|---|---|---|---|---|---|
| | $m^3\ m^{-3}$ | $m^3\ m^{-3}$ | $cm^{-1}$ | *Unitless* | $m^2$ | $W\ m^{-1}\ K^{-1}$ | $W\ m^{-1}\ K^{-1}$ |
| Upper peat | 0.80 | 0.08 | 0.02 | 0.3 | $5 \cdot 10^{-11}$ | 0.6 | 0.1 |
| Lower peat | 0.70 | 0.07 | 0.02 | 0.4 | $2 \cdot 10^{-12}$ | 0.75 | 0.12 |
| Mineral soil | 0.50 | 0.10 | $5.4 \cdot 10^{-4}$ | 0.19 | $2 \cdot 10^{-13}$ | 1.0 | 0.29 |
| Ice wedge | 0.99 | 0.00 | $5.4 \cdot 10^{-4}$ | 0.19 | 0.0 | 0.59 | 0.02 |




**Table 2: Minimum observed temperature at each rod and depth (°C)**

|  | a000 | a101 | a102 | a103 | a104 | a105 | a106 | a108 | a109 | Center Median | Rim Median |
|---|---|---|---|---|---|---|---|---|---|---|---|
| 10cm | -19.8 | -19.0 | -18.0 | -16.1 | -19.9 | -16.0 | -17.7 | -22.5 | -21.1 | -17.8 | -21.1 |
| 20cm | -18.4 | -18.3 | -17.2 | -15.7 | -19.2 | -15.1 | -19.1 | -20.8 | -19.9 | -17.1 | -19.7 |
| 30cm | -18.0 | -17.7 | -16.5 | -15.3 | -18.3 | -14.6 | -16.4 | -19.7 | -19.0 | -16.4 | -19.0 |
| 40cm | -17.2 | -16.8 | -15.7 | -14.3 | -17.6 | -14.0 | -15.6 | -18.7 | -18.0 | -15.7 | -18.0 |
| 50cm | -16.5 | -16.1 | -15.0 | -13.8 | -16.4 | -13.5 | -15.1 | -18.0 | -17.3 | -15.0 | -17.3 |



**Table 3: Freezing curtain duration at each rod and depth (days)**

|  | a000 | a101 | a102 | a103 | a104 | a105 | a106 | a108 | a109 | Center Median | Rim Median |
|---|---|---|---|---|---|---|---|---|---|---|---|
| 10cm | 29.6 | 37.5 | 38.5 | 42.9 | 37.6 | 38.5 | 41.6 | 28.3 | 29.5 | 38.5 | 29.5 |
| 20cm | 41.9 | 54.3 | 49.8 | 58.0 | 52.8 | 34.6 | 57.1 | 39.5 | 38.8 | 54.4 | 39.5 |
| 30cm | 57.0 | 59.8 | 57.1 | 65.0 | 59.4 | 62.1 | 64.1 | 50.8 | 52.5 | 60.9 | 52.5 |
| 40cm | 67.9 | 68.6 | 61.5 | 74.1 | 64.9 | 71.5 | 72.0 | 58.1 | 58.4 | 68.6 | 58.4 |
| 50cm | 68.0 | 77.1 | 68.5 | 80.8 | 75.6 | 81.8 | 77.4 | 64.1 | 65.0 | 77.3 | 65.0 |



**Table 4: Rank sum test results (*p*-values) for two hypotheses**

|  | 10cm | 20cm | 30cm | 40cm | 50cm |
|---|---|---|---|---|---|
| Rims experience colder minimum temperatures | 0.024 | 0.024 | 0.024 | 0.024 | 0.012 |
| Rims experience shorter freezing curtains | 0.012 | 0.012 | 0.012 | 0.024 | 0.012 |



**Table 5. Sensitivity analysis: minimum simulated temperature at top of ice wedge (°C)**

|  | rim -10cm | rim -5cm | rim +0cm | rim +5cm | rim +10cm | rim +15cm |
|---|---|---|---|---|---|---|
| center -0cm | -13.9 | -14.4 | -14.2 | -15.3 | -16.3 | -16.5 |
| center -10cm | -13.0 | -13.7 | -14.0 | -14.8 | -15.7 | -16.2 |
| center -20cm | -12.8 | -12.9 | -13.8 | -14.8 | -15.5 | -16.1 |
| center -30cm | -12.6 | -12.8 | -13.9 | -14.6 | -15.0 | -16.1 |
| center -40cm | -12.4 | -12.5 | -12.8 | -13.0 | -14.7 | -16.1 |



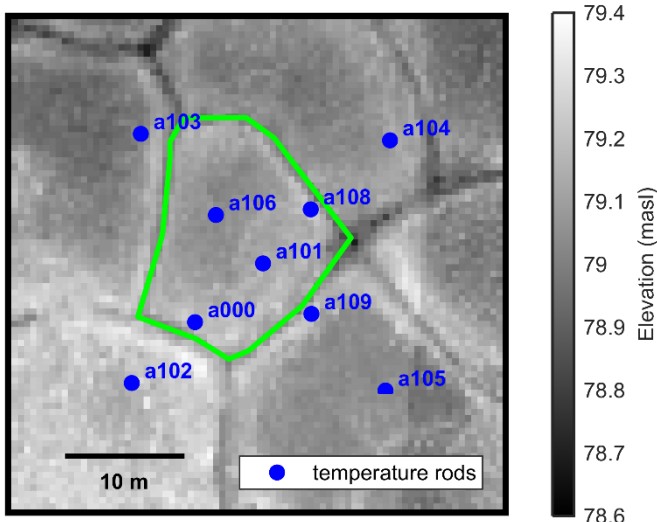

Figure 1. 50 cm resolution lidar DEM of field site.



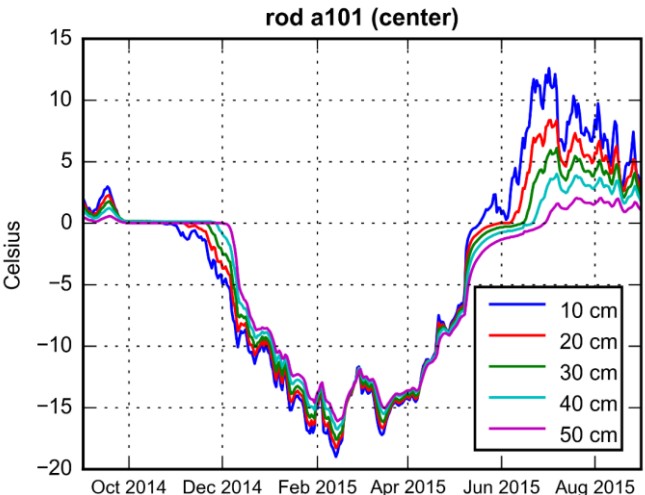

**Figure 2. Sample of observed temperature data from rod a101 (polygon center).**





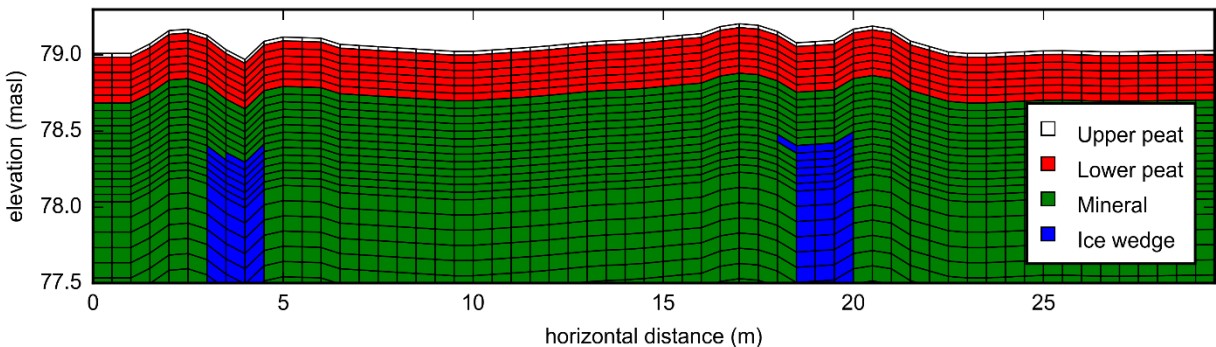

**Figure 3. Schematic of the 2D mesh of the field site (vertical exaggeration=4)**





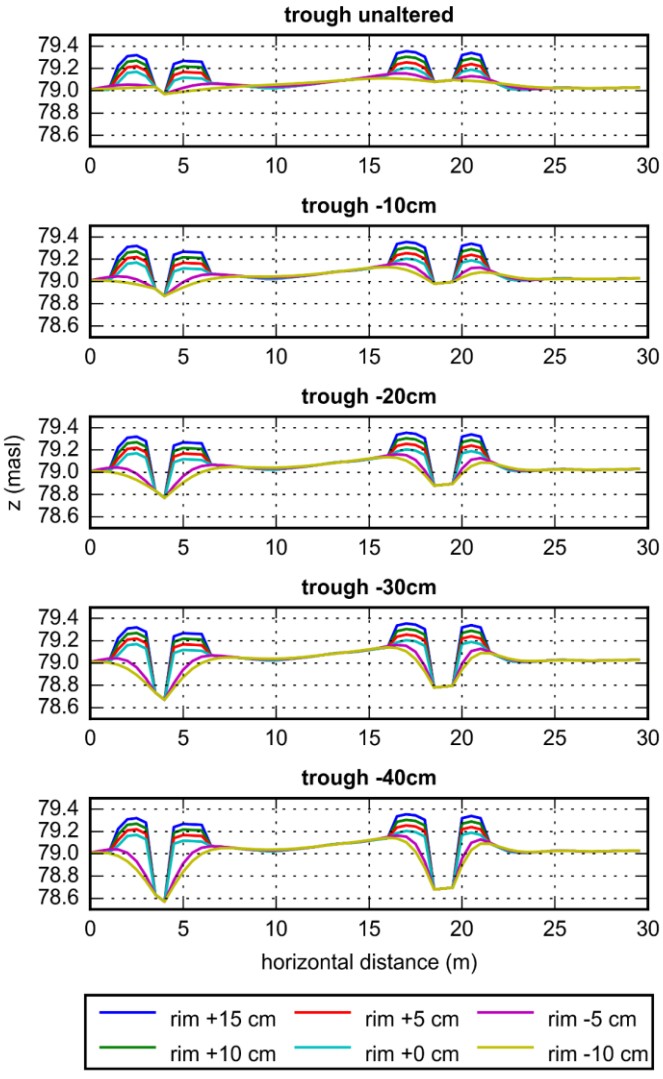

**Figure 4. Schematic illustrating the range of topographic conditions explored in the sensitivity analysis (vertical exaggeration=6)**



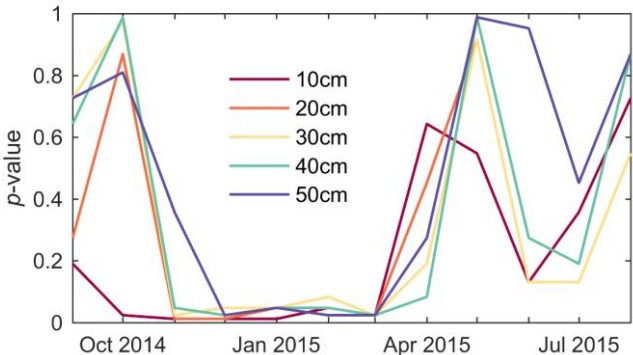

**Figure 5. Results of the rank sum test, evaluating whether mean monthly temperature are colder in the rims than in the centers.**



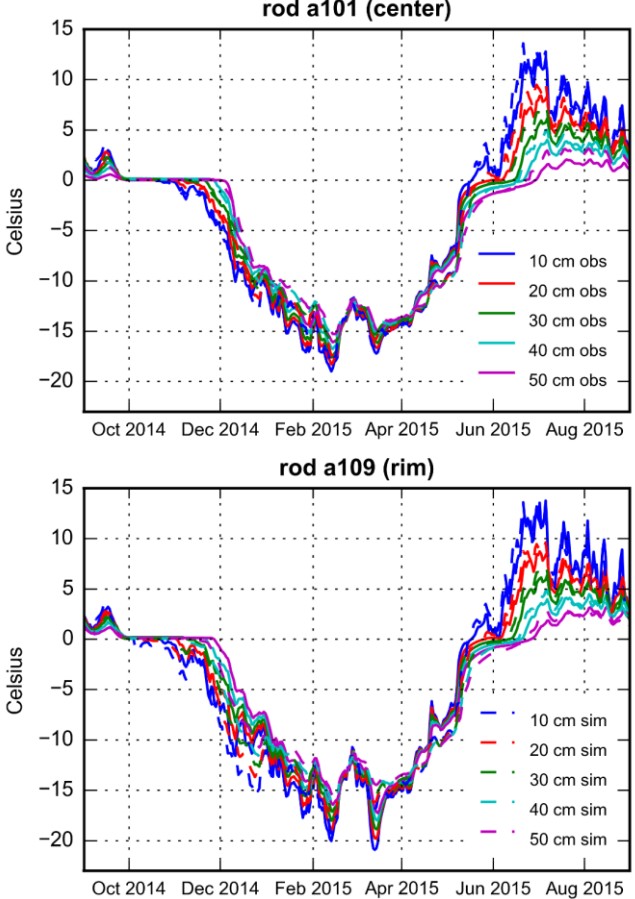

**Figure 6. Observed and simulated ground temperature from calibrated 2D simulation, at rods a101 (center) and a109 (rim)**



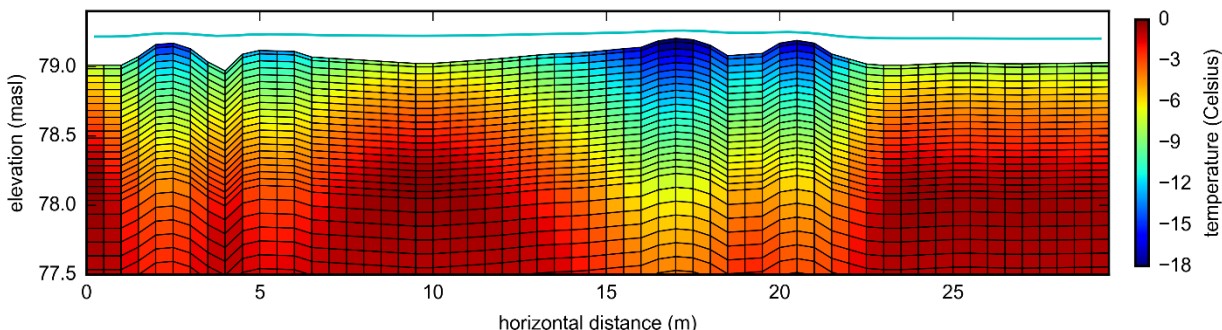

**Figure 7. Snapshot of simulated snowpack surface (cyan line) and ground temperature on December 24, 2014 (vertical exaggeration=4).**