# Peer review of "Microtopographic control on the ground thermal regime in ice wedge polygons"

_The Cryosphere, 2018_

## Referee Comment (RC1) · Anonymous Referee #1 · 7 Feb 2018

Overall a well written paper about the subsurface temperature dynamics in an ice wedge environment. The evaluation of snow depth effects on winter temperatures is relevant for the future of the tundra ecosystem. However the thermal contraction cracking is not only induced by the ice wedge temperature, but also by the temperature of the center of the polygon to supply the forces needed to pull the ice wedge apart. The wave of temperature change in time and depth causes the crack to form and propagate. It is therefore not clear whether the temperatures simulated in this study are relevant to the processes of cracking. The model however seems very capable of simulating this temperature dynamics and this by itself sufficient for this paper. Deep borehole data is referred to in the paper (this should however be Romanovsky's Franklin Bluffs site, not Clow for the region of your study) however the data are not used in the model to

show the bottom boundary is simulated correctly. It was very amusing to me that Noah "estimated land surface model output..." I think it should be NOAA?

Specific: P1L28 "~4 5" maybe "4 to 5" P9L5 "the" can be removed from "the phase change" Figure 2 and 6 need clarification, the lines are crowded.

---

## Author Comment (AC1) · 22 Feb 2018

Thank you very much for your feedback on our paper! After reading your concern about whether our results are relevant to ice wedge cracking, we revisited the literature on the topic. Below we present a more detailed review to contextualize our results, which we will include in the Background and Discussion sections of the next version of the manuscript.

The rule of thumb we cite – that cracking is possible when ground temperatures are below 13°C and cool at a rate of at least 0.1° d-1 for two days or more – comes from Morse and Burn (2013). A similar rule appears in Kokelj et al. (2014). In both cases, these conditions are presented as applying to the "top of the permafrost," without

specifying beneath which part of the polygon. These rules of thumb were established by reviewing a number of prior field investigations in which thermal conditions were monitored at the time of cracking. In some of these investigations, the focus was on temperatures at the top of the ice wedge (Christiansen, 2005; Allard and Kasper, 1998), while in others, temperature was monitored at the top of the permafrost beneath the polygon center (Fortier and Allard, 2005).

These studies, like all recent research on ice wedge cracking, are grounded in the foundational mechanical analysis of Lachenbruch (1962), who used a visco-elastic model of thermal strain in permafrost to determine that cracking is most likely when the permafrost is already cold, and the rate of cooling is rapid. Lachenbruch's analysis assumed uniform temperatures with depth, so that the horizontal stress at a point is a function of lithostatic pressure and the cooling history of the ground. The situation, as you point out, is more complicated if winter temperatures are non-uniform at the upper boundary of the permafrost. After an extensive literature search, we can find no mechanical analysis that quantifies the extent to which thermal contraction at the center of the polygon contributes to the stress experienced by the ice wedge when the wedge remains warmer. However, we can find several studies which either imply or demonstrate that alterations to thermal conditions at the periphery of the polygon are sufficient to change cracking behavior. For example:

- Mackay (1993) writes that the stresses that drive ice wedge cracking in low-centered polygons probably originate "spatially more in the area beneath the ridges, rather than in the areas beneath the polygon centres."

- Watanabe et al. (2017) write that ice wedge cracking activity at Svalbard is most common where "snow-free well-developed polygon rims further intensify the cooling of the active layer."

- Burn (2010) found that cracking in a previously inactive ice wedge was reinitiated after removing grasses that had been growing in the trough, thereby reducing snowpack

above the wedge.

- Burn (2004) also writes that, in general, "Troughs are sites of preferential snow accumulation, so that wedges beneath well-developed troughs are relatively warm and rarely crack."

- Finally, Lachenbruch (1962), in the conclusion of his mechanical analysis, suggests that ice wedge cracking may be suppressed "by increased winter snow accumulation in deepening interpolygonal troughs."

Our model takes this previous work as motivation for studying the sensitivity of permafrost temperature at the edge of the polygon to rim size and trough depth. Although we acknowledge that we cannot infer thermal stresses or probabilities of cracking from the results of our study, we use the previous research on ice wedge cracking as a precedent to forecast the overall effect on cracking activity as high-centered development effectively increases the insulation of ice wedges. In this light, we believe it is fair to project that thermokarst development likely has an inhibiting effect on cracking.

Regarding the temperature at the bottom boundary of the model, we will update our manuscript to cite Romanovsky's data set, which is located very close to our field site. The estimate of temperature at 50 m depth in Romanovsky's data set (-6$^\circ$C) varies slightly from the number we used from Clow's data set (-8$^\circ$C). When we updated the bottom boundary condition in our 2D simulation based on unaltered field site topography, the minimum winter temperature at the ice wedge increased slightly, by $\sim$0.3$^\circ$C. We are currently re-running our entire ensemble of simulations, and will update the results in the next version of the manuscript.

The Noah land surface model we refer to is used by NASA to estimate surface conditions in the Global Land Data Assimilation System (Rodell, 2004).

References

Allard M, Kasper JN. 1998. Temperature conditions for ice-wedge cracking: Field measurements from Salluit, northern Quebec. Proceedings of the 7th International Conference on Permafrost, pp. 5-12. National Research Council of Canada.

Burn CR. 2004. A field perspective on modelling 'single-ridge' ice-wedge polygons. Permafrost and Periglacial Processes 15, 59-65.

Burn CR. 2010. Experimental rejuvenation of ice-wedge cracking at Illisarvik, western Arctic coast, Canada. Presentation at American Geophysical Fall Meeting.

Christiansen HH. 2005. Thermal regime of ice-wedge cracking in Adventdale, Svalbard. Permafrost and Periglacial Processes 16, 87-98.

Fortier D, Allard M. 2005. Frost-cracking conditions, Bylot Island, eastern Canadian Arctic archipelago. Permafrost and Periglacial Processes 16, 145-161.

Kokelj SV, Lantz TC, Wolfe SA, Kanigan JC, Morse PD, Coutts R, Molina-Giraldo N, Burn CR. 2014. Distribution and activity of ice wedges across the forest-tundra transition, western Arctic Canada. Journal of Geophysical Research: Earth Surface 119, 2032-2047.

Lachenbruch AH. 1962. Mechanics of thermal contraction cracks and ice-wedge polygons in permafrost. Geological Society of America Special Paper, New York.

Mackay JR. 1993. Air temperature, snow cover, creep of frozen ground, and the time of ice-wedge cracking, western Arctic coast. Canadian Journal of Earth Sciences 30, 1720-1729.

Morse PD, Burn CR. 2013. Field observations of syngenetic ice wedge polygons, outer Mackenzie Delta, western Arctic coast, Canada. Journal of Geophysical Research: Earth Surface 118, 1320-1332.

Rodell M, et al. 2004. The Global Land Data Assimilation System. Bulletin of the American Meteorological Society 85, 381-394.

Watanabe T, Matsuoka N, Christiansen HH, Cable S. 2017. Soil physical and environmental conditions controlling patterned-ground variability at a continuous permafrost site, Svalbard. Permafrost and Periglacial Processes 28, 433-445.

---

## Referee Comment (RC2) · Anonymous Referee #2 · 16 Apr 2018

General comments:

The manuscript is well written and addresses an interesting topic of current relevance in the literature. The main conclusions support field evidence, conceptual models, and hypotheses from prior studies. The strength of this paper is that the authors were able to manipulate rim/trough morphology and examine their effect on the ground thermal regime explicitly using a numerical model. Perhaps the most interesting (and novel) conclusion was that varying rim height produced a greater change in minimum ice wedge temperature than varying trough depth. I think this point should be mentioned in the abstract.

I have 2 other general comments, which are further detailed in the next section (1) The authors presented two previous hypotheses regarding the formation of secondary

ice wedges (Burn and O'Neill 2015 and Dostovalov and Popov 1966). The discussion paragraph describing the influence of microtopography on thermal conditions could be strengthened by relating the results of the modelling to these competing views. This needn't be lengthy and would simply involve minor changes. See below.

(2) The interpretability of the tables and figures could be improved significantly by re-naming the field sites, and addressing some minor issues, particularly on Figure 6. See below.

Specific comments:

P. 2 Line 32. Suggest adding reference to recent paper describing temperature and cooling rate conditions during thermal contraction cracking: O'Neill H.B. and Christiansen H.H. Detection of ice‐wedge cracking in permafrost using miniature accelerometers. https://doi.org/10.1002/2017JF004343 Journal of Geophysical Research: Earth Surface.

P. 4 Lines 23-25. Is the sentence about thermokarst lakes needed? Nothing is mentioned about lakes later, and the thermal effect of lakes within 1 km likely have little to no effect on the temperatures at the depths measured in this paper.

P. 4. Line 32. Rods removed in what month, 2015?

P. 7 line 27. Can you quantify "comparing well"? What is the difference in the maximum snow depths?

P. 8. Line 1. I presume the RMSE is calculated on daily temperatures, but can you clarify please.

P. 8 Line 31. You hypothesized colder rims for your site, but this has been clearly demonstrated before. As this is the discussion section, I suggest referencing past studies that have explicitly addressed this (e.g., Christiansen 2005, Thermal Regime of Ice-wedge Cracking in Adventdalen, Svalbard. PPP DOI: 10.1002/ppp.523).

P. 9 Line 18. I'm not sure if your model program supports it, but the heat loss from the rims could be very nicely illustrated with a closeup of the rim/trough area showing ground heat flux vector arrows. Not necessary but would make a nice addition if easy to do.

P. 9 Line 21. This point about cooling at the wedge due to rim relief was explicitly mentioned in Christiansen (2005, reference above), and is stated in the conclusion: "Effective cooling of the active layer above the side of the ice wedges in the almost always snow-free ramparts permitted the top of the central part of the ice-wedge to attain the critical temperature of -15C. This appears to explain why thermal-contraction cracking is widespread even beneath snow-filled ice wedge troughs.". So, to say that it has not been emphasized in previous conceptual models is somewhat inaccurate. In light of this, you may wish to reword parts of the discussion accordingly.

P. 10 lines 7-16. "Regarding historical polygon development, the results provide evidence that feedbacks associated with..." I think this paragraph could do with more explicit reference to the two competing arguments by e.g., Burn and O'Neill (2015) and Dostovalov and Popov (1966). You set this up nicely in the background section, but the discussion paragraph would benefit by referencing Burn and O'Neill when you talk about primary wedge deactivation and secondary wedge cracking. E.g., line 10 could be reworded as something like "... the model results support the hypothesis of Burn and O'Neill (2015) that feedbacks associated with microtopographic change..."

Table 1. Typically both the frozen and unfrozen conductivity values are reported. Here you do not specify what you are reporting, but given the values I assume these are unfrozen? I suggest reporting both, as the difference is important. Also, may be worth clarifying what the "dry" thermal conductivity of an ice wedge means?!?

Table 2 and 3 (and Figure 1). I suggest renaming all of your instruments. The names a000 mean nothing to the reader, and make it difficult to determine which individual instrument is in the rim or center without looking at the map. I suggest renaming the

C1 to . . . and R1 to . . . for Center and Rim, respectively. This will make it much easier for the reader.

Figure 3. Suggest adding sentence to remind the reader that the simulation extends to 50 m depth and that only the upper ground is shown.

Figure 6. First, are the legend labels correct? Why are all either obs (a101) or sim (a109)?. Second, near impossible to differentiate the obs and sim. I suggest removing all but the 10 cm and 50 cm plots. We don't need to see all the inbetweeners, do we? This will make it possible to see the obs vs. sim.

Suggested technical corrections

P. 4 l. 11. "centered on a low-centered" consider revising (word repetition).

P. 8 l. 14. "was sufficient to make cracking favorable". Better as "was sufficient to favor cracking" ?

P. 9 l. 6 and 7. Suggest deleting "increasing the potential for pref. transmittance of heat". An increase in conductivity is exactly this, so text after the comma is redundant.

Table 4. Suggest changing "experience" to "with".

[Figure]

---

## Author Response (AR1)

**Response to Reviewer #1:**

**Thank you for your review of our paper, which we believe has led to a better numerical representation of our field site (by redefining the bottom boundary condition of our model to match Romanovsky's data), and a stronger discussion section (by including additional references addressing the impacts of rim height and trough depth on ice wedge cracking). Please find below our response to specific comments in bold. Additionally, we include a copy of the revised manuscript with changes tracked at the end of this document.**

Overall a well written paper about the subsurface temperature dynamics in an ice wedge environment. The evaluation of snow depth effects on winter temperatures is relevant for the future of the tundra ecosystem. However the thermal contraction cracking is not only induced by the ice wedge temperature, but also by the temperature of the center of the polygon to supply the forces needed to pull the ice wedge apart. The wave of temperature change in time and depth causes the crack to form and propagate. It is therefore not clear whether the temperatures simulated in this study are relevant to the processes of cracking. The model however seems very capable of simulating this temperature dynamics and this by itself sufficient for this paper.

**Thank you for your feedback on our manuscript! As we discussed in our previous author comment, we believe that a body of prior field work supports the idea that alterations to the ground thermal regime localized at the periphery of the polygon can alter ice wedge cracking behavior. We include citations for relevant studies (Mackay, 2000; Burn, 2004; Christiansen, 2005; Watanabe et al., 2017) in a new paragraph in our Discussion (page 9 lines 24-31 in the revised manuscript). We hope this provides better context for our interpretation of the model's results.**

Deep borehole data is referred to in the paper (this should however be Romanovsky's Franklin Bluffs site, not Clow for the region of your study) however the data are not used in the model to show the bottom boundary is simulated correctly.

**Thank you for directing us to borehole data closer to our site. As we mentioned in our previous comment, estimates of ground temperature at 50 m depth from Romanovsky's dataset (-6°C) vary slightly from Clow's dataset (-8°C). For the revised manuscript we have used Romanovsky's data as suggested, and re-run each of our simulations with the updated boundary condition. We also slightly changed our calibration of snow parameters (thermal conductivity of fresh snow = 0.021 W m-1 K-1 as opposed to 0.019; snow redistribution coefficient reduced 60% from model default as opposed to 90%), in order to maintain a low RMSE between observed and simulated temperature at our field site. Therefore, our results have changed slightly, but still support the conclusions of the previous manuscript.**

It was very amusing to me that Noah "estimated land surface model output..." I think it should be NOAA?

**As we mentioned in the author comment, the Noah land surface model is used by NASA's Global Land Data Assimilation System to estimate weather conditions globally.**

Specific:

P1L28 "~4 5" maybe "4 to 5"

**Thank you for catching this. We have made the change.**

P9L5 "the" can be removed from "the phase change"

**We have changed this wording as suggested.**

Figure 2 and 6 need clarification, the lines are crowded.

**We agree and have changed the size and formatting of the figures for improved clarity.**

**Response to Reviewer #2:**

**Thank you for your comments on our manuscript! We appreciate your suggestions for improvement, which we believe have increased the clarity of our findings (through edits to the abstract and figures) and helped provide a more complete and succinct summary of where our research fits within previous work on ground temperature and ice wedge cracking (through additional references in the Background and Discussion). Please find below our responses to specific comments in bold. We also include a revised version of the manuscript with changes tracked at the end of this document.**

General comments:

The manuscript is well written and addresses an interesting topic of current relevance in the literature. The main conclusions support field evidence, conceptual models, and hypotheses from prior studies. The strength of this paper is that the authors were able to manipulate rim/trough morphology and examine their effect on the ground thermal regime explicitly using a numerical model. Perhaps the most interesting (and novel) conclusion was that varying rim height produced a greater change in minimum ice wedge temperature than varying trough depth. I think this point should be mentioned in the abstract.

I have 2 other general comments, which are further detailed in the next section.

(1) The authors presented two previous hypotheses regarding the formation of secondary ice wedges (Burn and O'Neill 2015 and Dostovalov and Popov 1966). The discussion paragraph describing the influence of microtopography on thermal conditions could be strengthened by relating the results of the modelling to these competing views. This needn't be lengthy and would simply involve minor changes. See below.

(2) The interpretability of the tables and figures could be improved significantly by re-naming the field sites, and addressing some minor issues, particularly on Figure 6. See below.

**Thank you for your feedback on our manuscript! As suggested, we have modified the abstract to emphasize our finding that winter ice wedge temperature is more sensitive to rim height than to trough depth. We have also responded to your other two general comments, as described below.**

Specific comments:

P. 2 Line 32. Suggest adding reference to recent paper describing temperature and cooling rate conditions during thermal contraction cracking: O'Neill H.B. and Christiansen H.H. Detection of ice wedge cracking in permafrost using miniature accelerometers. https://doi.org/10.1002/2017JF004343 Journal of Geophysical Research: Earth Surface.

**Thank you. We have added this reference (page 3 line 3 in revision with markup).**

P. 4 Lines 23-25. Is the sentence about thermokarst lakes needed? Nothing is mentioned about lakes later, and the thermal effect of lakes within 1 km likely have little to no effect on the temperatures at the depths measured in this paper.

**We agree and have eliminated this sentence (page 4 lines 25-27).**

P. 4. Line 32. Rods removed in what month, 2015?

**We now specify September, 2015 (page 5 line 2).**

P. 7 line 27. Can you quantify "comparing well"? What is the difference in the maximum snow depths?

**The maximum observed snow depths at Deadhorse and Sagwon were 53 cm and 58 cm respectively, which are slightly higher than our simulated maximum snow depth of 45 cm. We now state this in the text on page 7, line 31.**

P. 8. Line 1. I presume the RMSE is calculated on daily temperatures, but can you clarify please.

**We now specify that RMSE is calculated on daily temperatures (page 8 line 4).**

P. 8 Line 31. You hypothesized colder rims for your site, but this has been clearly demonstrated before. As this is the discussion section, I suggest referencing past studies that have explicitly addressed this (e.g., Christiansen 2005, Thermal Regime of Ice-wedge Cracking in Adventdalen, Svalbard. PPP DOI: 10.1002/ppp.523).

**Thank you. We have included a citation of Christiansen, 2005 (page 9 line 2).**

P. 9 Line 18. I'm not sure if your model program supports it, but the heat loss from the rims could be very nicely illustrated with a closeup of the rim/trough area showing ground heat flux vector arrows. Not necessary but would make a nice addition if easy to do.

**Thank you for this suggestion. We appreciate the idea and looked into producing this figure, but were unable to do so with the available model output.**

P. 9 Line 21. This point about cooling at the wedge due to rim relief was explicitly mentioned in Christiansen (2005, reference above), and is stated in the conclusion: "Effective cooling of the active layer above the side of the ice wedges in the almost always snow-free ramparts permitted the top of the central part of the ice-wedge to attain the critical temperature of -15C. This appears to explain why thermal-contraction cracking is widespread even beneath snow-filled ice wedge troughs." So, to say that it has not been emphasized in previous conceptual models is somewhat inaccurate. In light of this, you may wish to reword parts of the discussion accordingly.

**We agree and have eliminated the sentence beginning with "The high sensitivity of ice wedge temperature to rim height has not been emphasized…" We also have included a new paragraph in the Discussion section that better contextualizes our results with a number of field studies, including the paper by Christiansen (page 9 lines 24-33).**

P. 10 lines 7-16. "Regarding historical polygon development, the results provide evidence that feedbacks associated with..." I think this paragraph could do with more explicit reference to the two competing arguments by e.g., Burn and O'Neill (2015) and Dostovalov and Popov (1966). You set this up nicely in the background section, but the discussion paragraph would benefit by referencing Burn and O'Neill when you talk about primary wedge deactivation and secondary wedge cracking. E.g., line 10 could be reworded as something like "...the model results support

the hypothesis of Burn and O'Neill (2015) that feedbacks associated with microtopographic change..."

**We have reworded the paragraph as suggested. The sentence in the middle of the paragraph now reads "Regarding historical polygon development, the results support the hypothesis of Burn and O'Neill (2015) that feedbacks associated with microtopographic change are sufficient to explain the presence of secondary wedges in modern and ancient polygons…" (page 10 line 22).**

Table 1. Typically both the frozen and unfrozen conductivity values are reported. Here you do not specify what you are reporting, but given the values I assume these are unfrozen? I suggest reporting both, as the difference is important. Also, may be worth clarifying what the "dry" thermal conductivity of an ice wedge means?!?

**We have updated the table to include frozen saturated thermal conductivity, which the model estimates as a function of unfrozen saturated thermal conductivity and porosity. In the model input files, we specified 0.02 W m-1 K-1, or approximately the thermal conductivity of air, as the "dry" thermal conductivity of an ice wedge. This number was essentially meaningless, however, as the ice wedge never became less than fully saturated. To avoid confusion, we have replaced this number with 'N/A'. We thank the reviewer for pointing this out so that we could make this clarification and avoid confusing readers.**

Table 2 and 3 (and Figure 1). I suggest renaming all of your instruments. The names a000 mean nothing to the reader, and make it difficult to determine which individual instrument is in the rim or center without looking at the map. I suggest renaming for Center and Rim, respectively. This will make it much easier for the reader.

**Thank you for this suggestion. We agree that renaming the sensors to indicate rim or center location is a far more logical system. We have replaced the old sensor codes with new names throughout the figures, tables, and text.**

Figure 3. Suggest adding sentence to remind the reader that the simulation extends to 50 m depth and that only the upper ground is shown.

**We have clarified this point both in Figure 3 and Figure 7.**

Figure 6. First, are the legend labels correct? Why are all either obs (a101) or sim (a109)?. Second, near impossible to differentiate the obs and sim. I suggest removing all but the 10 cm and 50 cm plots. We don't need to see all the inbetweeners, do we? This will make it possible to see the obs vs. sim.

**We have eliminated the plots of temperature at 20-40 cm depth as suggested, and changed the figure size and formatting to improve clarity. The figure is more focused and clear now. We thank the review for suggesting these modifications that make the figure more informative and interpretable.**

Suggested technical corrections:

P. 4 l. 11. "centered on a low-centered" consider revising (word repetition).

**We have changed to "surrounds a low-centered" (page 4 line 13).**

P. 8 l. 14. "was sufficient to make cracking favorable". Better as "was sufficient to favor cracking"?

**We have changed the wording as suggested (page 8 line 17).**

P. 9 l. 6 and 7. Suggest deleting "increasing the potential for pref. transmittance of heat". An increase in conductivity is exactly this, so text after the comma is redundant.

**We have deleted the suggested text (page 9 line 9).**

Table 4. Suggest changing "experience" to "with".

**We have changed this wording.**

[revised manuscript text omitted]

---

## Editor Decision (ED1)

[revised manuscript text omitted]

| Insert a co     | lumn title   |           |           |                  |                    |                   |                   |                   |
|-----------------|--------------|-----------|-----------|------------------|--------------------|-------------------|-------------------|-------------------|
| such as "G      | Ground       |           | 1         | 1                | I                  | I                 | 1                 | 1                 |
| materal"        |              | Residual  | Van       | Van              | Intrinsic          | Saturated         | Saturated         | Dry thermal       |
|                 | Porosity     | water     | Genuchten | Genuchten        | permeability       | (thawed) thermal  | (frozen) thermal  | conductivity      |
| $\leftarrow$    | 1            | content   | α         | m                | 1 2                | conductivity      | conductivity      | 5                 |
|                 | $m^3 m^{-3}$ | $m^3 m^3$ | cm⁻¹      | unitless         | $m^2$              | $W m^{-1} K^{-1}$ | $W m^{-1} K^{-1}$ | $W m^{-1} K^{-1}$ |
| Upper peat      | 0.80         | 0.08      | 0.02      | 0.3              | 5.10-11            | 0.6               | 1.8               | 0.1               |
| Include sym     | bol as       |           |           |                  |                    |                   |                   |                   |
| per Table S     | 1            | 0.07      | 0.02      | <mark>0.4</mark> | $2 \cdot 10^{-12}$ | 0.75              | 1.9               | 0.12              |
| pear            |              |           |           |                  |                    |                   |                   |                   |
| Mineral
soil | 0.50         | 0.10      | 5.4.10-4  | 0.19             | $2 \cdot 10^{-13}$ | 1.0               | 2.0               | 0.29              |
| Ice wedge       | 0.99         | 0.00      | 5.4.10-4  | 0.19             | 0.0                | 0.59              | 2.2               | N/A               |
| Include         | symbol a     | s         | I         | I                | l                  | l                 | l                 | I                 |
| per Tab         | ole S1       |           |           |                  |                    |                   |                   |                   |
|                 |              |           |           |                  |                    |                   |                   |                   |

**Table 2: Minimum observed temperature at each rod and depth (°C)**

|             | con1           | con?  | con3  | con/  | con5  | conf  | rim1   | rim?   | rim3  | Center | Rim    |
|-------------|----------------|-------|-------|-------|-------|-------|--------|--------|-------|--------|--------|
| 1           | cent           | Cell2 | cens  | Cell4 | cens  | Cello | 111111 | 111112 | 11113 | Median | Median |
| 10 c | m -19.0        | -17.7 | -16.1 | -16.0 | -18.0 | -19.9 | -21.1  | -19.8  | -22.5 | -17.8  | -21.1  |
| 20e         | m -18.3 | -19.1 | -15.7 | -15.1 | -17.2 | -19.2 | -19.9  | -18.4  | -20.8 | -17.1  | -19.7  |
| 30 c | m -17.7        | -16.4 | -15.3 | -14.6 | -16.5 | -18.3 | -19.0  | -18.0  | -19.7 | -16.4  | -19.0  |
| 40 c | m -16.8 | -15.6 | -14.3 | -14.0 | -15.7 | -17.6 | -18.0  | -17.2  | -18.7 | -15.7  | -18.0  |
| 50e         | m -16.1 | -15.1 | -13.8 | -13.5 | -15.0 | -16.4 | -17.3  | -16.5  | -18.0 | -15.0  | -17.3  |
|             | I              | I     | I     | 1     | I     | I     | I      | I      | I     | I      |        |

Depth

**Table 3: Freezing curtain duration at each rod and depth (days)**

|   | •                | cen1 | cen2 | cen3 | cen4 | cen5 | cen6 | rim1 | rim2 | rim3 | Center
Median | Rim
Median |
|---|------------------|------|------|------|------|------|------|------|------|------|------------------|---------------|
|   | $\uparrow$       |      |      |      |      |      |      |      |      |      | Wiedian          | Wiedian       |
| 1 | l0 <del>cm</del> | 37.5 | 41.6 | 42.9 | 38.5 | 38.5 | 37.6 | 29.5 | 29.6 | 28.3 | 38.5             | 29.5          |
| 2 | 20 <del>cm</del> | 54.3 | 57.1 | 58.0 | 34.6 | 49.8 | 52.8 | 38.8 | 41.9 | 39.5 | 54.4             | 39.5          |
|   | 0cm              | 59.8 | 64.1 | 65.0 | 62.1 | 57.1 | 59.4 | 52.5 | 57.0 | 50.8 | 60.9             | 52.5          |
| 1 | 40 <del>cm</del> | 68.6 | 72.0 | 74.1 | 71.5 | 61.5 | 64.9 | 58.4 | 67.9 | 58.1 | 68.6             | 58.4          |
| 4 | 50 <del>cm</del> | 77.1 | 77.4 | 80.8 | 81.8 | 68.5 | 75.6 | 65.0 | 68.0 | 64.1 | 77.3             | 65.0          |

\_Depth

Table 4: Rank sum test results (*p*-values) for two hypotheses

| $\wedge$                              | 10 <del>cm</del> | 20 <del>cm</del>                            | 30 <del>cm</del> | 40 <del>cm</del> | 50 <del>cm</del> |  |  |  |  |  |  |
|---------------------------------------|------------------|---------------------------------------------|------------------|------------------|------------------|--|--|--|--|--|--|
| Rims with colder minimum temperatures | 0.024            | 0.024                                       | 0.024            | 0.024            | 0.012            |  |  |  |  |  |  |
| Rims with shorter freezing curtains   | 0.012            | 0.012                                       | 0.012            | 0.024            | 0.012            |  |  |  |  |  |  |
|                                       |                  |                                             |                  |                  |                  |  |  |  |  |  |  |
|                                       |                  |                                             |                  |                  |                  |  |  |  |  |  |  |
| Hypothesis                            | A
a
c      | dd a header
cross the de
alled "Depth | pths
"        |                  |                  |  |  |  |  |  |  |

Table 5. Sensitivity analysis: minimum simulated temperature at top of ice wedge (°C)

|   | 1                                    | <del>rim</del> -10 <del>cm</del> | <del>rim -5cm</del> | r <del>im</del> +0 <del>cm</del> | rim-+5cm | rim-+10cm | rim-+15em |
|---|--------------------------------------|----------------------------------|---------------------|----------------------------------|----------|-----------|-----------|
|   | center -0cm                          | -13.94                           | -14.08              | -14.37                           | -14.56   | -14.92    | -15.16    |
|   | c <del>enter</del> -10 <del>cm</del> | -13.18                           | -13.42              | -14.07                           | -14.39   | -14.87    | -14.98    |
|   | <del>center</del> -20 <del>cm</del>  | -12.88                           | -13.33              | -13.85                           | -14.13   | -14.57    | -15.00    |
|   | center30cm                           | -12.99                           | -13.11              | -13.52                           | -13.99   | -14.34    | -14.71    |
|   | center -40cm                         | -12.87                           | -12.98              | -13.38                           | -13.89   | -14.28    | -14.57    |
|   |                                      |                                  |                     |                                  |          |           |           |
| 5 |                                      |                                  |                     |                                  |          |           |           |

Add a title here, "Trough/center manipulation (cm)

Add a header across the column titled: "Rim manipulation (cm)"

Figure 1. 50 cm resolution lidar DEM of field site.

Add a location map to this figure, and probably partition to Fig. 1a and 1b.

Figure 2. Sample of observed temperature data from rod a101 (polygon center).

---

## Author Response (AR2)

**Response to Editor:**

Thank you very much for your markup and comments on our manuscript. Please find attached our revised manuscript with tracked changes. At each location where a suggestion was made for a minor editorial change (*e.g.*, replacing 'topography' with 'microtopography') we have followed the suggestion. The most significant additions to content are in the third paragraph of Section 2 and the fifth paragraph of Section 5.2, where we have expanded on our discussion of secondary ice wedge formation by contrasting epigenetic wedges (such as those at our field site) with syngenetic and anti-syngenetic wedges (in which networks secondary wedges are uncommon). We appreciate your referral to relevant literature and suggestion to include these concepts in our manuscript, which makes our analysis of the model's implications for interpreting secondary ice wedge networks more succinct and impactful. We also now cite Morse and Burn's paper on frost blister formation in low centered polygons at the suggested locations in our manuscript. The paper is highly relevant to our investigation and we underscore the compatibility between its results and our own in the second paragraph of Section 5.1.

We have also reformatted our figures, tables, and supplemental information as suggested. Specific changes include:

- Axis labels in all figures now begin with capital letters.
- Figure 1 now includes a map displaying the location of the study site in Alaska.
- The weight of the lines in Figure 6 has been reduced.
- Tables 1 and S1 both include parameter names (e.g., dry thermal conductivity) and symbols (e.g., $\lambda_u$).
- GLDAS and ATS are both defined and cited in Text S1.
- A list of references is included at the end of the Supplement.

Thank you for suggesting these changes to increase the clarity of our manuscript.

Sincerely,

Chuck Abolt

[revised manuscript text omitted]